# c-Jun N-terminal kinase (JNK) signaling contributes to cystic burden in polycystic kidney disease

**Abigail O. Smith**[1], **Julie A. Jonassen**[2], **Kenley M. Preval**[1], **Roger J. Davis**[1], **Gregory J. Pazour**[1] *

**1** Program in Molecular Medicine, University of Massachusetts Chan Medical School, Biotech II, Worcester, Massachusetts, United States of America, **2** Department of Microbiology and Physiological Systems, University of Massachusetts Chan Medical School, Worcester Massachusetts, United States of America

* gregory.pazour@umassmed.edu

**Data Availability Statement:** All relevant data are within the manuscript and its Supporting Information files.

## Abstract

Polycystic kidney disease is an inherited degenerative disease in which the uriniferous tubules are replaced by expanding fluid-filled cysts that ultimately destroy organ function. Autosomal dominant polycystic kidney disease (ADPKD) is the most common form, afflicting approximately 1 in 1,000 people. It primarily is caused by mutations in the transmembrane proteins polycystin-1 (Pkd1) and polycystin-2 (Pkd2). The most proximal effects of *Pkd* mutations leading to cyst formation are not known, but pro-proliferative signaling must be involved for the tubule epithelial cells to increase in number over time. The c-Jun N-terminal kinase (JNK) pathway promotes proliferation and is activated in acute and chronic kidney diseases. Using a mouse model of cystic kidney disease caused by *Pkd2* loss, we observe JNK activation in cystic kidneys and observe increased nuclear phospho c-Jun in cystic epithelium. Genetic removal of *Jnk1* and *Jnk2* suppresses the nuclear accumulation of phospho c-Jun, reduces proliferation and reduces the severity of cystic disease. While *Jnk1* and *Jnk2* are thought to have largely overlapping functions, we find that *Jnk1* loss is nearly as effective as the double loss of *Jnk1* and *Jnk2*. Jnk pathway inhibitors are in development for neurodegeneration, cancer, and fibrotic diseases. Our work suggests that the JNK pathway should be explored as a therapeutic target for ADPKD.

## Author summary

Autosomal dominant polycystic kidney disease is a leading cause of end stage renal disease requiring dialysis or kidney transplant. During disease development, the cells lining the kidney tubules proliferate. This proliferation transforms normally small diameter tubules into fluid-filled cysts that enlarge with time, eventually destroying all kidney function. Despite decades of research, polycystic kidney disease remains incurable. Furthermore, the precise signaling events involved in cyst initiation and growth remain unclear. The c-Jun N-terminal kinase (JNK), is a major pathway regulating cellular proliferation and differentiation but its importance to polycystic kidney disease was not known. We show that

**Funding:** This work was supported by National Institute of Health (www.nih.gov) grants DK103632 to G.J.P and DK107220 to R.J.D. The funders had no role in study design, data collection and analysis, decision to publish, or preparation of the manuscript.

**Competing interests:** The authors have declared that no competing interests exist.

JNK activity is elevated in cystic kidneys and that reducing JNK activity decreases cyst growth pointing to JNK inhibition as a therapeutic strategy for treating polycystic kidney disease.

## Introduction

Autosomal dominant polycystic kidney disease (ADPKD) is the most common form of inherited kidney disease, afflicting approximately 1 in 1,000 people in the United States and worldwide. Patients with ADPKD exhibit gradual kidney function decline due to uncontrolled epithelial cell proliferation and secretion that transforms narrow uriniferous tubules into large, fluid-filled cysts. The majority of ADPKD cases are due to mutations in either of two transmembrane proteins, polycystin-1 (Pkd1) and polycystin-2 (Pkd2), that form a heterotetrameric complex in the primary ciliary membranes [1] It is widely believed that perturbing this ciliary complex, either by loss-of-function mutations or disrupting cilia structure triggers the cellular phenotype that leads to cyst formation [2–4] Although the precise mechanism by which the polycystin complex preserves tubule architecture remains obscure, some aspects of pro-cystic signaling have been established. For example, Pkd1 or Pkd2 loss leads to reduced intracellular calcium and, subsequently, an abnormal cellular response to cyclic adenosine monophosphate (cAMP) levels [5–7] In mutant epithelial cells, elevated cAMP promotes increased fluid secretion and epithelial cell proliferation [8] cAMP reduction via vasopressin 2 receptor antagonism is the mechanism of action of tolvaptan, the single FDA-approved drug for patients with ADPKD [9,10] Unfortunately, tolvaptan slows but does not halt disease progression and is not appropriate for all ADPKD patients due to side effects [11] To improve treatments for ADPKD, we must search for alternative pro-cystic signaling pathways.

Prior studies showed that the c-Jun N-terminal kinase (JNK) signaling pathway is activated in cells overexpressing exogenous Pkd1 [12,13] or Pkd2 [14] constructs. A later study found the opposite, that Pkd1 loss activated JNK signaling, while Pkd1 overexpression repressed JNK activity [15] Reports of JNK activity in cystic tissues are also conflicting [16,17], and no followup studies established JNK's role in cyst formation. JNK is a member of the MAP kinase family, which also includes Erk1/2, p38, and Erk5. JNK pathway activators include extracellular stimuli such as UV irradiation, osmotic stress, and cytokines that initiate an intracellular phosphorylation cascade through upstream MAP kinase kinase kinases (MAP3K). Jnk-associated MAP3Ks converge on two MAP kinase kinases (MAP2Ks), Mkk4 and Mkk7. The MAP2Ks phosphorylate MAP kinases including the Jnk paralogs: Jnk1 (Mapk8), Jnk2 (Mapk9), and Jnk3 (Mapk10). Jnk1 and Jnk2 are ubiquitously expressed, while Jnk3 expression is restricted primarily to the central nervous system and testis [18] Although Jnks have a wide array of substrates, the most studied are the activator protein-1 (AP-1) transcription factors, particularly c-Jun, for which the pathway is named. Increased AP-1 levels have been detected in cystic kidneys in humans and mice [16] Furthermore, AP-1 promotes proliferation and cell survival by regulating oncogene transcription [19] including *c-Myc*, which was recently shown to contribute directly to cystic kidney disease [20,21]

JNK activation has been detected in many forms of kidney disease [22] In animal models, JNK inhibition prior to ischemia-reperfusion or tubule obstruction reduces inflammation and fibrosis, and preserves kidney function [23–27] Interestingly, acute kidney injury exacerbates polycystic kidney disease [28–30] In chronic kidney insult, progressive interstitial fibrosis contributes to organ failure. JNK inhibition reduces pro-fibrotic factors in the kidney [26,27]

Furthermore, researchers produced severe kidney fibrosis in mice by overexpressing the JNK target *c-Jun* [31]

This study aimed to investigate the role of JNK signaling in ADPKD using *in vivo* models to genetically perturb the pathway. Here we show that *Pkd2* deletion increases JNK activation, which contributes to cystic kidneys in young animals and cystic liver in older animals. Jnk1 is more important to the phenotype development than is Jnk2. Overall, our results encourage further investigation of the JNK pathway as a novel therapeutic candidate for treating ADPKD.

## Methods

### Ethics statement

These studies were approved by the Institutional Animal Care and Use Committee of the University of Massachusetts Chan Medical School Protocol 201900265 (A-1174).

### Mouse studies

The following mouse strains have been described previously: $Pkd2^{fl}$ [32], $Jnk1^{fl}$ [33], $Jnk2^{null}$ [34], $Rosa26\text{-}Cre^{ERT2}$ [35], $Ask1^{-/-}$ [36], $Mlk2^{-/-}$ [37] and $Mlk3^{-/-}$ [38]. The *Ask1* (B6.129S4-Map3k5<tm1Hijo>) mouse was provided by the RIKEN BRC through the National Bio-Resource Project of MEXT, Japan. *Stra8-iCre* [39] was used to convert $Jnk1^{fl}$ to $Jnk1^{null}$. All mice were C57BL/6J congenics maintained by backcrossing to C57BL/6J purchased from Jackson Laboratory (Bar Harbor Maine, USA).

For juvenile onset disease model, mothers were dosed with tamoxifen (200 mg/kg) by oral gavage on postnatal days 2, 3, and 4. The pups remained with nursing mothers until euthanasia at postnatal day 21. For the adult-onset disease model, animals were treated with tamoxifen (50 mg/kg) by intraperitoneal injection on postnatal days 21, 22, and 23. Mice were euthanized 24 weeks after first injection. Both sexes were used in all studies.

### Histology

Tissues were fixed by immersion overnight in 10% formalin (Electron Microscopy Sciences) in phosphate-buffered saline and then embedded in paraffin. Sections were deparaffinized and stained with hematoxylin and eosin (H&E) or one-step trichrome. Images of stained sections were obtained with a Zeiss Axio Scan.Z1 slide scanner with brightfield capabilities using the 20X objective. Cystic index was calculated using ImageJ software to outline kidney sections, apply a mask to differentiate cystic from non-cystic regions, and measure the two-dimensional areas. Cystic index = cystic area/total kidney area x 100%.

For immunofluorescent staining, sections were deparaffinized, antigens were retrieved by autoclaving for 30 min in 10 mM sodium citrate, pH 6.0 and stained with primary antibodies diluted in TBST (10 mM Tris, pH 7.5, 167 mM NaCl, and 0.05% Tween 20) plus 0.1% cold water fish skin gelatin (Sigma-Aldrich). Alexa Fluor–labeled secondary antibodies (Invitrogen) were used to detect the primary antibodies. Primary antibodies used included aquaporin 2 (1:100; Sigma # 5200110), phospho S63 c-Jun (1:1000, Cell Signaling Technology), phospho S10 histone H3 (1:250; Millipore # 06570), SMA (1:50,000, Sigma # A5228), collagen (1:250 Abcam ab260043). FITC-conjugated lectins were added with secondary antibodies: Lotus tetragonolobus agglutinin (LTA, 1:50, Vector Labs) and Dolichos biflorus agglutinin (DBA, 1:20, Vector Labs). Nuclei were labeled with 4′,6-diamidino-2-phenylindole (DAPI). Fluorescent images were obtained with a Zeiss LSM900+ Airyscan microscope. Fluorescent slide scans were obtained using Zeiss Axio Scan.Z1 slide scanner.

## Gene expression

Kidneys were stored at -80C in RNAlater (Qiagen) until RNA was isolated. For total RNA isolation, tissues were homogenized using TissueLyser II (Qiagen) and RNA isolated using the RNeasy Mini Kit (Qiagen). cDNA was synthesized using SuperScript II Reverse Transcription (Invitrogen). Real time quantitative PCR was performed with KAPA SYBR FAST Universal reagent (Roche) using an Eppendorf Realplex2 cycler. All qPCR reactions were performed in triplicate and melting curves verified that a single product was amplified. Standard curves were generated by 5-fold serial dilutions of a pool of untreated mouse kidney cDNA, and for each gene, the threshold cycle was related to log cDNA dilution by linear regression analysis. Gene expression data were normalized to glyceraldehyde-3-phosphate dehydrogenase expression. The following primers were used: *Pkd2 forward* CGAGGAGGAGGATGACGAAGAC; *Pkd2 reverse* TGGAAACGATGCTGCCAATGGA; *Gapdh forward* GCAATGCATCCTGCA CCACCA; *Gapdh reverse* TTCCAGAGGGGGCCATCCACA.

## Immunoblotting

For Fig 1 only, kidneys stored in RNAlater were homogenized in Buffer RLT (RNeasy MiniKit, Qiagen). Protein was precipitated from the supernatant by adding 9 volumes of 100% methanol, collected by centrifugation at 3,000xg for 10 min at 4C followed by three washes in 90% methanol. Protein pellets were reconstituted in 2X SDS-PAGE loading buffer. In all other experiments, frozen kidneys were homogenized in ice-cold RIPA buffer (150 mM NaCl, 1% Triton X-100, 0.05% sodium deoxycholate, 0.01% SDS, 50 mM Tris-HCl, pH 7.5) supplemented with Complete Mini EDTA-free Protease Inhibitor cocktail tablets (Roche), sodium orthovanadate (0.5 mM), sodium fluoride (10 mM), and phenylmethylsulfonyl fluoride (1 mM). Equal amounts of protein were loaded and separated in 12% SDS-PAGE gels and transferred to Immobilon-FL PVDF membranes (Millipore). Membranes were blocked for 1 hour at room temperature with Intercept (TBS) Blocking Buffer (Li-Cor) or 5% non-fat dry milk in TBST followed by incubation with primary antibodies overnight at 4C.

The following antibodies were used: JNK1/2 (1:1000, BD Pharmingen # 554285), c-Jun (1:200, Santa Cruz Biotechnology # 74543), phospho T183/Y185 SAPK/JNK (1:1000, Cell Signaling Technology # 9251), phospho S63 c-Jun (1:1000, Cell Signaling Technology # 9261), glyceraldehyde-3-phosphate dehydrogenase (1:10,000, Proteintech, 60004-1-Ig), SMA (1:1,000, Sigma # A5228), glyceraldehyde-3-phosphate dehydrogenase (1:1,000, Cell Signaling Technology # 3683S), phospho T180/Y182 p38, (Cell Signaling Technology # 9211). Primary antibodies were detected using near infrared secondary antibodies (Li-Cor) and blots were imaged on Odyssey Li-Cor imager. Quantification was performed using Image Studio Lite software.

## Results

### Postnatal deletion of *Pkd2* activates JNK signaling in juvenile mouse kidneys

To assess JNK signaling in an *in vivo* ADPKD model, we induced *Pkd2* deletion in postnatal mice by the tamoxifen inducible RosaCre$^{ERT2}$ driver. This widely expressed Cre causes gene deletion in most cells, including kidney tubule epithelium as well as immune and other cells of the kidney [40,41] We administered tamoxifen via maternal oral gavage on postnatal (P) days 2–4 and harvested on P21. Treating *Pkd2 flox* (*fl*) homozygotes (*Rosa26-Cre$^{ERT2}$; Pkd2$^{fl/fl}$*) resulted in extensive kidney cysts. In contrast, heterozygous mice (*Rosa26-Cre$^{ERT2}$; Pkd2$^{fl/+}$*) similarly treated, exhibited no cysts and thus served as controls (Fig 1A).

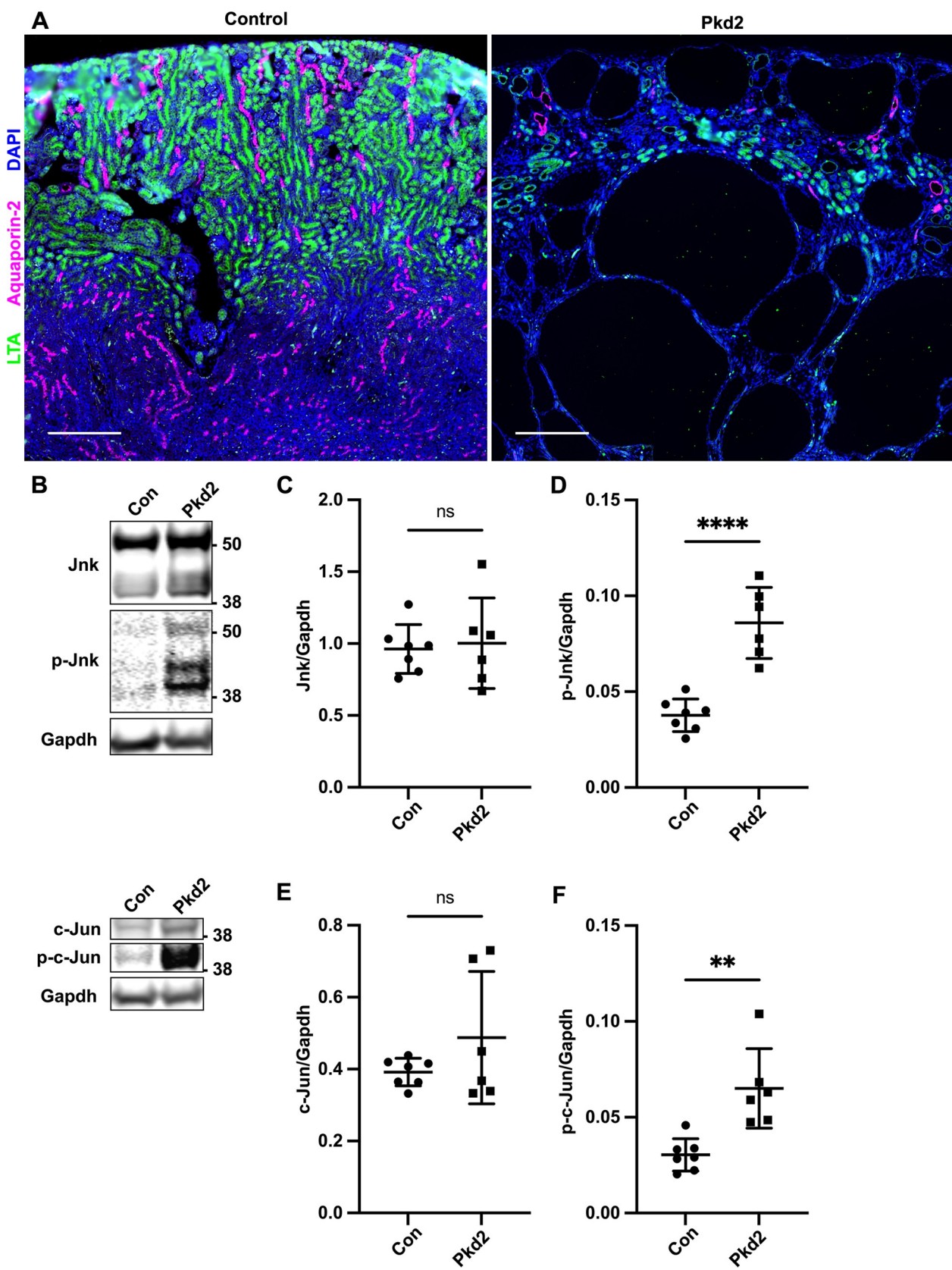

**Fig 1. Postnatal deletion of Pkd2 activates JNK signaling in juvenile mouse kidneys.** Mice with the following genotypes were treated with tamoxifen by maternal transfer at P2-4 and collected at P21: Con (Rosa26-CreERT2; Pkd2fl/+), Pkd2 (Rosa26-CreERT2; Pkd2fl/fl). (A) Kidney sections were probed for tubule epithelial markers LTA (proximal tubules) and aquaporin-2 (collecting ducts). Nuclei were marked by DAPI. Images are slide scans obtained on Zeiss Axio Scan.Z1 with 20X objective. Scale bar is 500 microns. (B) Whole kidney protein samples were immunoblotted for total Jnk, phospho T183/Y185 Jnk, total c-Jun, phospho S63 c-Jun, and loading control Gapdh. (C-F) Quantification of immunoblots described in (B). N is 7 (Con), 6 (Pkd2). ****, P < 0.0001; **, P < 0.01; ns, not significant by unpaired two-tailed t-test. Error bars indicate SD.

Total Jnk protein in *Pkd2* mutant kidneys was unchanged compared to controls but phosphorylated Jnk was significantly increased (Fig 1B–1D). These antibodies recognize both Jnk1 and Jnk2, each of which is alternatively spliced to generate 54 kDa and 46 kDa products (S3A Fig). Pkd2 loss did not significantly alter total c-Jun levels, but did elevate phosphorylated c-Jun (Fig 1B, 1E and 1F). Our findings indicate that *Pkd2* deletion activates JNK signaling and could drive cyst formation.

## JNK inhibition reduces severity of cystic phenotype in juvenile *Pkd2* mutant mice

Our finding that the loss of Pkd2 activates JNK signaling could indicate that JNK activation drives cyst formation, or cyst formation could activate JNK signaling. To distinguish these possibilities, we tested how JNK inhibition affects cyst formation driven by Pkd2 loss. Mice express three Jnk paralogs. *Jnk1* and *Jnk2* are widely expressed including in kidney, while *Jnk3* is limited to brain and testis [18] Thus, we focused on *Jnk1* and *Jnk2*. Losing both genes causes embryonic lethality. To circumvent lethality, we intercrossed parents carrying germline *Jnk2* deletions and floxed *Jnk1* alleles with *Pkd2$^{fl}$; Rosa26-Cre$^{ERT2}$* alleles used previously. Offspring, carrying assorted alleles, were treated with tamoxifen by maternal transfer on P2-4, harvested on P21, and genotyped (Fig 2). *Pkd2* heterozygotes carrying any number of wild-type *Jnk* alleles had normal kidney to body weight and no evidence of structural abnormalities or cysts. Kidneys lacking *Pkd2* but carrying at least one wild-type allele of *Jnk1* and *Jnk2* showed severe cystic disease similar to *Pkd2* deletion alone. Kidneys lacking *Pkd2* and all functional *Jnk* alleles had a 23% reduction in two-kidney to body weight and a 16% reduction in cystic index (Fig 2B and 2C). Hematoxylin and eosin (H&E) stained mid-sagittal sections revealed large cysts at the cortical-medullary boundary and smaller cysts in the cortex and medulla in *Pkd2* mutants with intact JNK activity. *Jnk* deletion reduced cysts in the cortex and medulla. Cysts remained at the cortical-medullary boundary but were less extensive (Fig 2A). Importantly, Jnk inactivation had no significant effect on kidney architecture, two-kidney to body weight or cystic index of *Pkd2* heterozygotes in the time period we examined (Fig 2A–2C). Our observation that *Jnk* deletion reduces disease severity supports the hypothesis that JNK activation contributes to cystic disease.

We observed variability in kidney to body weight in *Pkd2* mutants (Fig 2B). We hypothesized that this was due to variation in *Pkd2* levels after Cre-mediated deletion. To test our hypothesis, we measured *Pkd2* mRNA levels by RT-qPCR and normalized to *Gapdh*. Non-cystic controls (*Rosa26-Cre$^{ERT2}$; Pkd2$^{fl/+}$*) used throughout this study, are expected to have about one half as much *Pkd2* message as Cre-negative animals and this was observed (S1A Fig). Cystic groups (*Rosa26-Cre$^{ERT2}$; Pkd2$^{fl/fl}$*) with or without JNK activity showed reduced *Pkd2* mRNA levels compared to non-cystic controls but the reduction did not reach significance due to variation between animals. Importantly, we found similar variation and no difference in mean *Pkd2* mRNA levels in *Pkd2* mutants with and without Jnk alleles. This finding indicates that the difference in cystic burden between *Pkd2* mutants with and without JNK activity is not due to systematic differences in *Pkd2* levels. Plotting cystic burden vs. Pkd2 mRNA levels

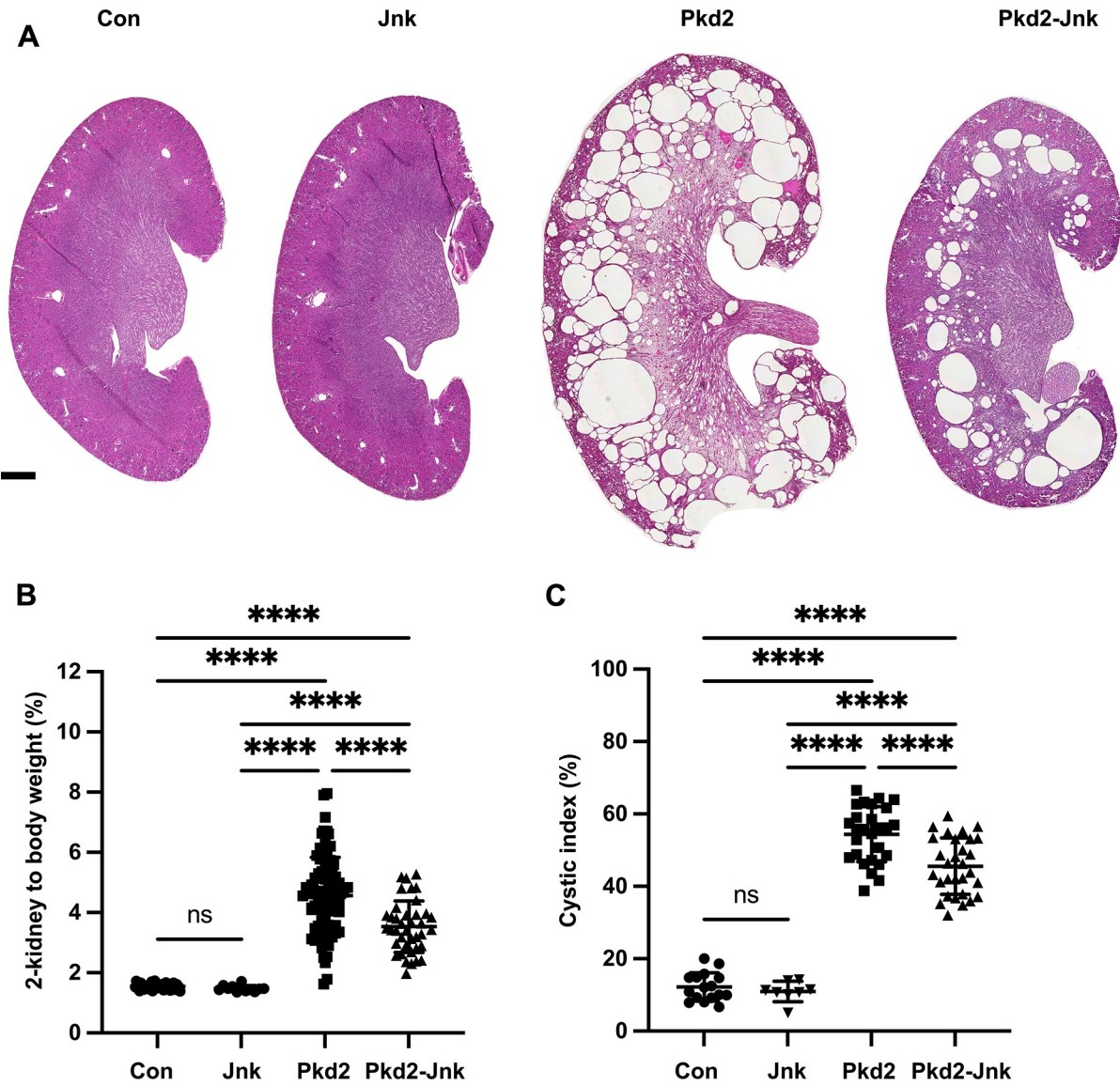

**Fig 2. JNK inhibition reduces kidney cysts in juvenile Pkd2 mutant mice.** Mice with the following genotypes were treated with tamoxifen by maternal transfer at P2-4 and collected at P21: Con (Rosa26-CreERT2; Pkd2fl/+), Jnk (Rosa26-CreERT2; Pkd2fl/+; Jnk1fl/fl; Jnk2null/null), Pkd2 (Rosa26-CreERT2; Pkd2fl/fl; Jnk1+/+, fl/+; Jnk2+/+, +/-), and Pkd2-Jnk (Rosa26-CreERT2; Pkd2fl/fl; Jnk1fl/fl; Jnk2null/null). (A) Kidney sections from P21 mice were stained with H&E to show the extent of disrupted organ architecture. Scale bar is 500 microns and applies to all images in the panel. (B) Cystic burden was quantified using the ratio of 2-kidney weight / body weight x 100%. N is 42 (Con), 12 (Jnk), 88 (Pkd2), 39 (Pkd2-Jnk). ****, P < 0.0001 by one-way ANOVA followed by Tukey multiple comparison test with multiplicity-adjusted p-values. Error bars indicate SD. (C) Cystic index (cystic area / total kidney area x 100%) was calculated for mid-sagittal H&E-stained kidney sections. N is 16 (Con), 8 (Jnk), 27 (Pkd2), 30 (Pkd2-Jnk). ****, P < 0.0001 by one-way ANOVA followed by Tukey multiple comparison test with multiplicity-adjusted p-values. Error bars indicate SD.

for individual kidney samples shows that for a given Pkd2 level, Jnk-deleted mutants tend to have a lower cystic burden than mutants with intact Jnk signaling (S1B Fig). However, this relationship did not reach statistical significance given the variation in Pkd2 levels in both groups.

Multiple upstream MAP3Ks activate JNK. Identifying and inhibiting relevant MAP3Ks in the context of cystic kidney disease could be therapeutically beneficial. To this end, we selected

three MAP3K genes, *Ask1 (Map3k5)*, *Mlk2 (Map3k10)*, and *Mlk3 (Map3k11)*, with connections to kidney disease for further analysis. *Ask1* inhibition reduces kidney and liver fibrosis [42] Cdc42 and Rac1 mediate JNK activation in the context of polycystin overexpression in cells [12,14] and activate *Mlk2* and *Mlk3* [37,43,44] However, in our model, *Ask1* deletion (S2A Fig) or double deletion of *Mlk2* and *Mlk3* (S2B Fig) did not reduce cystic burden in *Pkd2* mutants. Thus, identifying relevant MAP3Ks will require further investigation.

## *Pkd2* deletion activates the transcription factor c-Jun in kidney tubule epithelial cells

The c-Jun subunit of the AP-1 transcription factor complex is dually phosphorylated by Jnk on serines 63 and 73 [45] To detect nuclear-localized phosphorylated c-Jun, indicating JNK activation, we probed kidney sections for phospho S63 c-Jun. In control kidneys, we found no positive nuclei in proximal tubule cells (defined by LTA staining) or collecting duct cells (defined by DBA staining) (Fig 3). Cystic kidneys due to *Pkd2* loss exhibited extensive phospho S63 c-Jun nuclear staining in both proximal tubules and collecting ducts. Consistent with most cysts in this model deriving from collecting ducts, we observed more phospho S63 c-Jun positive cells in DBA-positive tubules compared to LTA-positive tubules. *Jnk* deletion reduced phospho S63 c-Jun positive cells nearly to control levels (Fig 3). Our findings confirm that JNK activation occurs in the tubule epithelium and correlates positively with regions of increased cyst formation.

## JNK inhibition reduces tubule epithelial cell proliferation in juvenile *Pkd2* mutant mice

Cyst growth depends on tubule epithelial cell proliferation, and a recent study found differential expression of cell cycle genes in *Pkd2* mutant mouse kidneys [46] To determine whether *Jnk* deletion reduces tubule cell proliferation in *Pkd2* mutant kidneys, we probed tissue sections for the mitotic marker phospho S10 histone H3 (Fig 4). As expected, *Pkd2* mutant kidneys showed markedly increased proliferation compared to controls. Proximal tubule cell proliferation nearly doubled from 2.3% in control kidneys to 4.3% in *Pkd2* kidneys. *Jnk* deletion returned the rate to control levels (Fig 4B). In collecting ducts, proliferation was nearly ten times higher in *Pkd2* mutants compared to controls and *Jnk* deletion reduced proliferation by 43% (Fig 4C). These findings suggest that JNK inhibition reduces cyst formation by inhibiting tubule epithelial cell proliferation.

## JNK inhibition reduces fibrosis in juvenile *Pkd2* mutant mice

JNK signaling promotes interstitial fibrosis in non-cystic kidney disease models [25,26] As fibrosis also contributes to advanced cystic kidney disease [47], we hypothesized that *Jnk* deletion may reduce fibrosis in *Pkd2* mutants. To measure fibrosis, we stained kidney sections for alpha-smooth muscle actin (SMA), a marker of active myofibroblasts (Fig 5). As expected, SMA staining was restricted to perivascular regions in control kidneys. In contrast, *Pkd2* mutants exhibited significant SMA expression surrounding cystic and non-cystic tubules while *Pkd2* mutants lacking JNK activity showed reduced SMA staining (Fig 5A) and SMA protein levels (Fig 5C and 5D). It is possible that fibrosis is reduced in Jnk-deleted mutants secondary to their reduction in cystic burden. To better understand this, we plotted cystic index vs. SMA intensity (Fig 5B). As expected, SMA intensity correlated positively with cystic index. However, the slope of the relationship was diminished in *Pkd2* mutants that lacked JNK activity. While the difference between the two slopes did not reach statistical significance

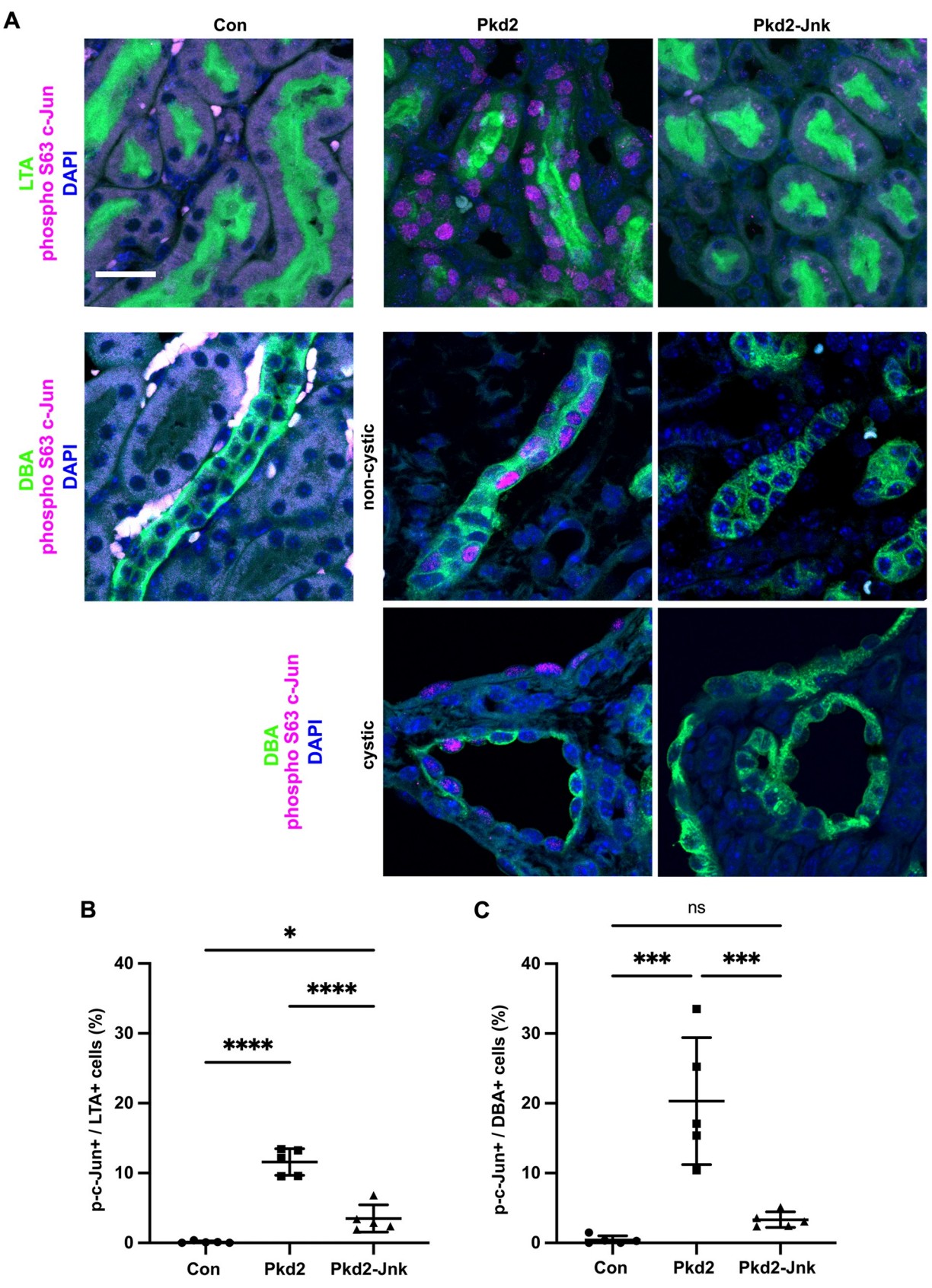

**Fig 3. Pkd2 deletion activates c-Jun in kidney tubule epithelial cells.** Mice with the following genotypes were treated with tamoxifen by maternal transfer at P2-4 and collected at P21: Con (Rosa26-CreERT2; Pkd2fl/+), Pkd2 (Rosa26-CreERT2; Pkd2fl/fl; Jnk1+/+, fl/+; Jnk2+/+, +/-), and Pkd2-Jnk (Rosa26-CreERT2; Pkd2fl/fl; Jnk1fl/fl; Jnk2null/null). (A) Kidney sections were probed for phospho S63 c-Jun and tubule epithelial markers LTA (proximal tubules) or DBA (collecting ducts). Examples of non-cystic and cystic collecting ducts are shown. Nuclei were marked by DAPI. Images are maximum projection of z-stacks (20 slices at 0.5um intervals) obtained on Zeiss LSM 900 Airyscan microscope with 40X objective. Scale bar is 20 microns and applies to all images in the panel. (B-C) Quantification of the proportion of tubule epithelial cells with nuclei positive for phospho S63 c-Jun. (B) Proximal tubules (LTA+) and (C) collecting ducts (DBA+) were quantified separately. N is 5 animals per group, 1000–2000 cells per animal. ****, P < 0.0001; ***, P< 0.001; *, P < 0.05; ns, not significant by one-way ANOVA followed by Tukey multiple comparison test with multiplicity-adjusted p-values. Error bars indicate SD.

(P = 0.055), the trend suggests that JNK inhibition affects myofibroblast activation independently from cystic burden.

## Jnk1 is primarily responsible for reducing cystic disease in juvenile *Pkd2* mutant kidneys

*Jnk1* and *Jnk2* are largely redundant in development but their roles diverge in adult tissues in a complex manner that is incompletely understood, yet critical for developing JNK inhibitor therapies [48,49] To determine how each *Jnk* gene contributes to cysts, we crossed parents carrying *Pkd2^fl^; Rosa26-Cre^ERT2^* alleles and germline *Jnk2^null^* or *Jnk1^null^* alleles. We treated mice as in Fig 2. Collagen staining with trichrome and anti-collagen 1A revealed that *Pkd2* heterozygotes were non-cystic with normal collagen deposition (Fig 6A). In contrast, *Pkd2* mutants contained large cysts at the cortical-medullary boundary and smaller cysts throughout. Collagen deposits were notable at the outer medulla and cyst boundaries. As shown in Fig 2, complete JNK deletion reduced cortical-medullary cysts. Collagen staining was visible but reduced. *Pkd2* mutants lacking only *Jnk1* exhibited fewer cysts and less fibrosis than *Pkd2* mutants with intact JNK. In contrast, *Pkd2* mutants lacking only *Jnk2* resembled cystic *Pkd2* mutants with JNK signaling intact. Kidney to body weight ratios and cystic indices support the histological evidence that *Jnk1* deletion reduces cysts more than *Jnk2* deletion (Fig 6B and 6C).

Immunoblots revealed increased c-Jun phosphorylation in *Pkd2* mutants (Fig 6D and 6E). Combined *Jnk1* and *Jnk2* deletion reduced phospho S63 c-Jun to near control levels. *Jnk1* deletion also reduced phospho S63 c-Jun to near control levels while *Jnk2* deletion did not. Our findings complement a recent study in which *Jnk1* deletion, but not *Jnk2* deletion, limited ischemia-reperfusion injury in mouse kidneys [23]

To measure Jnk isoform expression and phosphorylation, we probed immunoblots with antibodies against total and phosphorylated Jnk (S3A Fig). Two independent splicing events generate four isoforms each from Jnk1 and Jnk2. One site regulates inclusion of mutually exclusive exon 7a/7b, however these forms are not distinguishable by immunoblot. The other dictates choice of a C-terminal coding exon yielding long (p54) and short (p46) isoforms of both *Jnk1* and *Jnk2*, which can be distinguished by immunoblot (S3A Fig). Dual phosphorylation at Thr-Pro-Tyr activates all Jnk isoforms [50,51], and we detected increased phosphorylation of p54 and p46 in *Pkd2* mutant kidneys (S3A and S3C Fig). *Jnk2* deletion alone reduced p54 and phospho-p54 as effectively as total *Jnk* deletion. However, *Jnk1* deletion had no effect, suggesting p54 derives mostly from Jnk2. Total p46 was reduced by the double loss of both Jnk isoforms and by the single loss of Jnk2 but was not greatly affected by the loss of Jnk1. In contrast, phospho-p46 was reduced by the double loss and by the loss of Jnk1 suggesting that most of the phosphorylated form of p46 derives from Jnk1 even though Jnk2 produces most of this isoform. We detected a third band (p45), whose phosphorylation was elevated in all *Pkd2* mutants including those lacking Jnk1 or Jnk2. The observation that p45 remains in both *Jnk1* and *Jnk2* mutants suggests that it can be generated from either gene perhaps through a yet undescribed splicing event or a post translational modification. Alternatively, p45 could arise

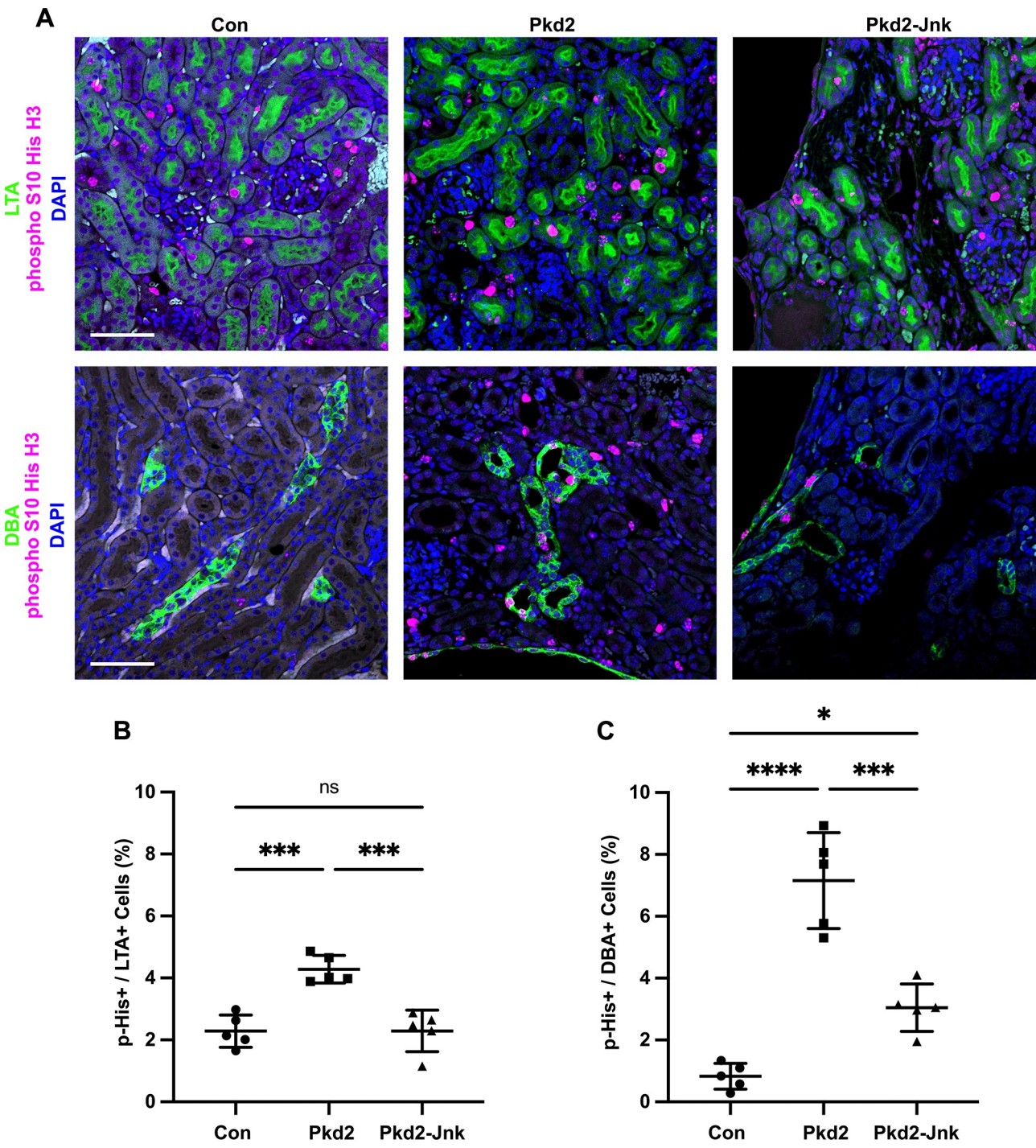

**Fig 4. JNK inhibition reduces tubule cell proliferation in juvenile Pkd2 mutant mice.** Mice with the following genotypes were treated with tamoxifen by maternal transfer at P2-4 and collected at P21: Con (Rosa26-CreERT2; Pkd2fl/+), Pkd2 (Rosa26-CreERT2; Pkd2fl/fl; Jnk1+/+, fl/+; Jnk2+/+, +/-), and Pkd2-Jnk (Rosa26-CreERT2; Pkd2fl/fl; Jnk1fl/fl; Jnk2null/null). (A) Kidney sections from P21 mice were probed for the mitotic marker phospho S10 histone H3 along with tubule epithelial markers LTA (proximal tubules) and DBA (collecting ducts). Nuclei were marked with DAPI. Images are maximum projection of z-stacks (10 slices at 0.5um intervals) obtained on Zeiss LSM 900 Airyscan microscope with 20X objective. Scale bar is 50 microns and applies to all images in the panel. (B-C) Quantification of the proportion of tubule cells with nuclei positive for phospho S10 histone H3. (B) Proximal tubule cells (LTA+) and (C) collecting duct cells (DBA+) were quantified separately. N is 5 animals per group, 1000–2000 cells per animal. ****, P < 0.0001; ***, P< 0.001; *, P < 0.05; ns, not significant by one-way ANOVA followed by Tukey multiple comparison test with multiplicity-adjusted p-values. Error bars indicate SD.

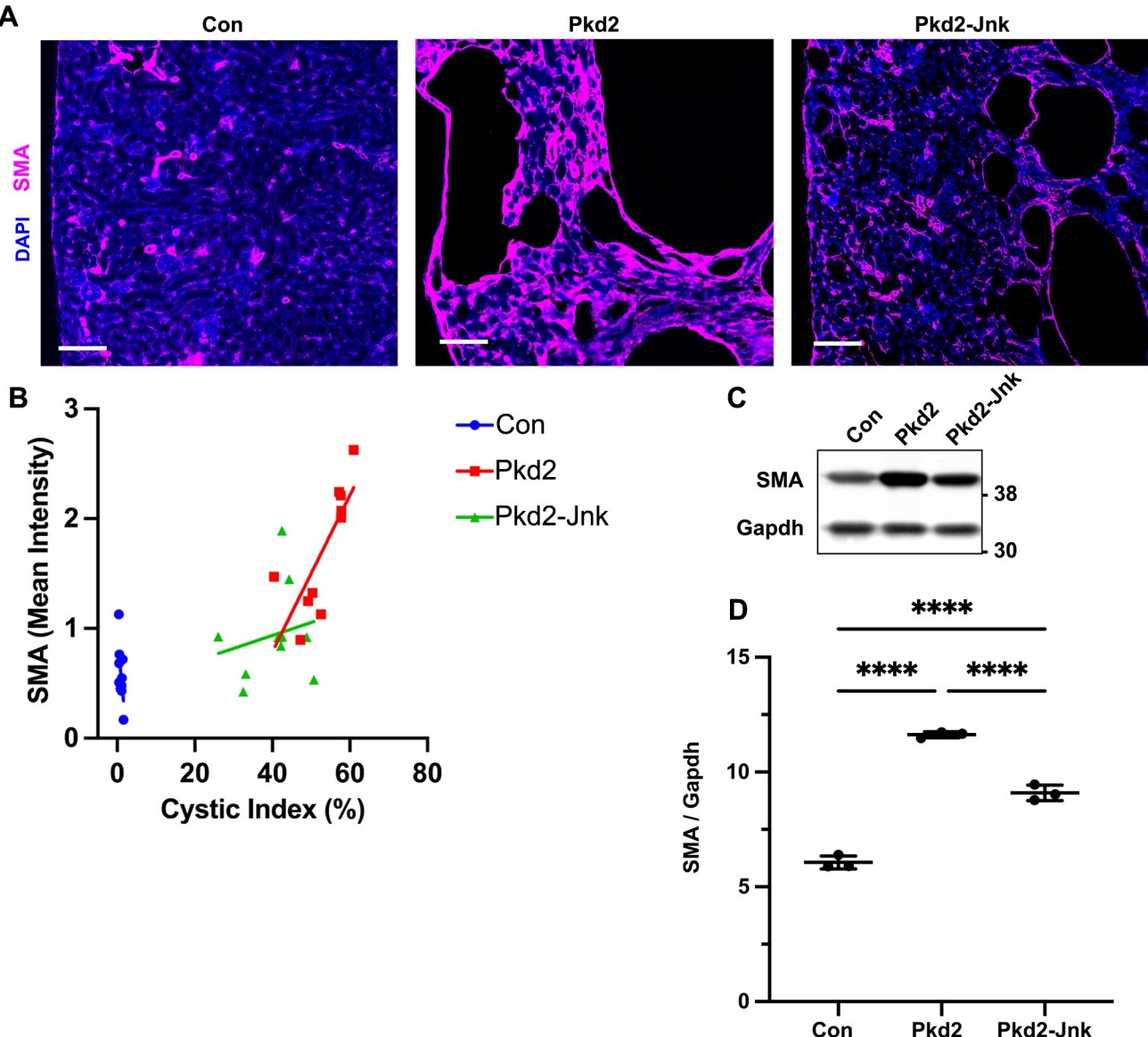

**Fig 5. JNK inhibition reduces fibrosis in juvenile Pkd2 mutant mice.** Mice with the following genotypes were treated with tamoxifen by maternal transfer at P2-4 and collected at P21: Con (Rosa26-CreERT2; Pkd2fl/+), Pkd2 (Rosa26-CreERT2; Pkd2fl/fl; Jnk1+/+, fl/+; Jnk2+/+, +/-), and Pkd2-Jnk (Rosa26-CreERT2; Pkd2fl/fl; Jnk1fl/fl; Jnk2null/null). (A) Kidney sections from P21 mice were probed for SMA, a marker of active fibroblasts. Nuclei were stained with DAPI. Images were obtained on Zeiss Axio Scan.Z1 with 20X objective. Scale bar is 100 microns. (B) SMA intensity was compared to cystic index for 10 animals per group. SMA mean intensity was measured using ImageJ software and divided by non-cystic area (mean intensity / non-cystic area x 100,000). Cystic index (cystic area / total area x 100%) was determined for DAPI staining using Image J software. Linear regression analysis shows that the difference between slopes for Pkd2 ($y = 0.07x-2$) and Pkd2-Jnk ($y = 0.01x+0.5$) approaches but does not reach significance ($P = 0.055$). (C) Whole kidney protein samples from P21 mice were immunoblotted for SMA and loading control Gapdh. (D) Quantification of immunoblots described in (C). N is 3 animals per groups. ****, $P < 0.0001$ by one-way ANOVA followed by Tukey multiple comparison test with multiplicity-adjusted p-values. Error bars indicate SD.

from antibody cross-reactivity with a different protein. Possibly, MAP kinase p38, as p38's phosphorylation pattern matches p45 (S3D Fig) however, p38 migrates faster than p45 making this unlikely. Our results show that *Pkd2* loss induces phosphorylation of long and short isoforms of Jnk1 and Jnk2 with the loss of phospho-Jnk1 p46 correlating best with disease protection.

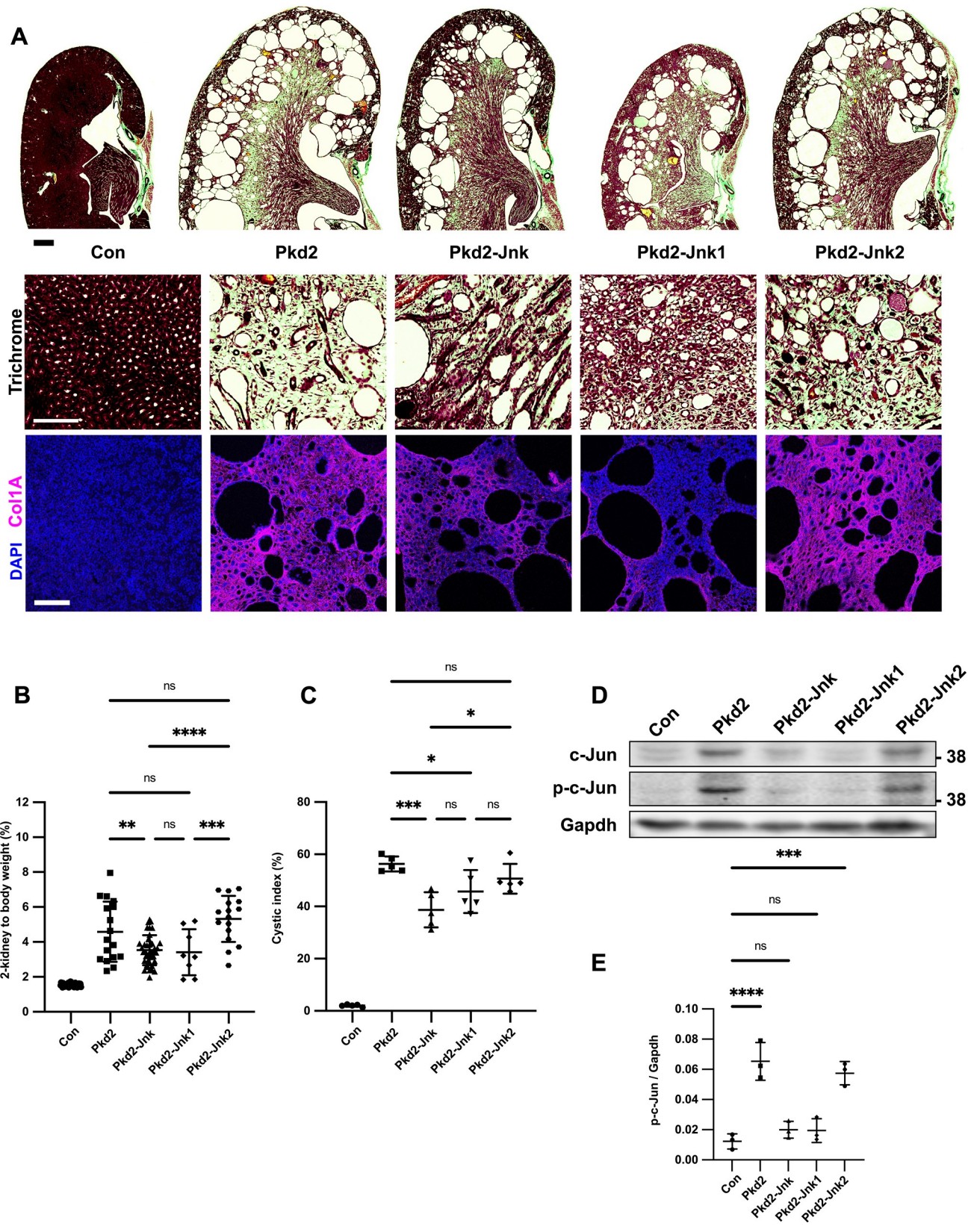

**Fig 6. Jnk1 deletion rather than Jnk2 is primarily responsible for reducing cystic disease in juvenile Pkd2 mutant kidneys.** Mice with the following genotypes were treated with tamoxifen by maternal transfer at P2-4 and collected at P21: Pkd2-Jnk1 (Rosa26-CreERT2; Pkd2fl/fl; Jnk1null/null; Jnk2 +/+); Pkd2-Jnk2 (Rosa26-CreERT2; Pkd2fl/fl; JNK1+/+; Jnk2null/null). We compared the results from these animals to the groups described previously: Con (Rosa26-CreERT2; Pkd2fl/+; JNK1+/+; Jnk2+/+), Pkd2 (Rosa26-CreERT2; Pkd2fl/fl; JNK1+/+; Jnk2+/+), Pkd2-Jnk (Rosa26-CreERT2; Pkd2fl/fl; Jnk1fl/fl; Jnk2null/null). (A) Kidney sections were stained with one-step trichrome which marks collagen fibers pale green, cytoplasm red, and nuclei dark blue. Scale bar for full size kidney scans is 500 microns. Trichrome insets show detail from inner medulla. Scale bar is 100 microns and applies to all images in the row. Additional kidney sections were probed using anti-collagen 1 antibody. Nuclei were marked by DAPI. Fluorescent images are maximum projection of z-stacks (6 slices at 0.5um intervals) obtained on Zeiss LSM 900 Airyscan microscope with 10X objective. Scale bar is 100 microns and applies to all images in the row. (B) Cystic burden was quantified using the ratio of 2-kidney weight / body weight x 100%. Con and Pkd2-Jnk data is the same as shown in Fig 2B. Pkd2 data overlaps with Fig 2B, but only includes animals with four wild-type alleles of Jnk. N is 42 (Con), 17 (Pkd2), 39 (Pkd2-Jnk), 8 (Pkd2-Jnk1), 16 (Pkd2-Jnk2). ****, P < 0.0001; ***, P< 0.001; **, P < 0.01; ns, not significant by one-way ANOVA followed by Tukey multiple comparison test with multiplicity-adjusted p-values. Error bars indicate SD. (C) Cystic index (cystic area / total kidney area * 100%) was calculated for mid-sagittal trichrome-stained kidney sections. N is 5 animals per group. ***, P< 0.001; *, P < 0.05; ns, not significant by one-way ANOVA followed by Tukey multiple comparison test with multiplicity-adjusted p-values. Error bars indicate SD. (D) Whole kidney protein lysates were immunoblotted for total c-Jun, phospho S63 c-Jun and loading control Gapdh. (E) Quantification of immunoblots described in (D). N is 3 animals per group. ****, P < 0.0001; ***, P< 0.001; ns, not significant by one-way ANOVA followed by Tukey multiple comparison test with multiplicity-adjusted p-values. Error bars indicate SD.

## JNK deletion reduces severity of cystic liver disease in adult *Pkd2* mutant mice

Inhibiting JNK signaling reduced cystic kidney disease in a rapidly progressing model of ADPKD. However, humans with ADPKD accumulate cysts over decades. Thus, we wanted to evaluate JNK signaling in a slowly progressing disease model. In mice, timing of cystic gene deletion determines rate of disease progression. *Pkd1* loss prior to P13 causes cyst accumulation within weeks, while deletion after P14 delays cysts for 5–6 months [52,53] For our slow-progressing model, we delivered tamoxifen at P21-23. *Pkd2* mutants aged 6 months showed no signs of kidney cysts in histological sections or by 2-kidney to body weight ratios (S4 Fig). Segregation by sex revealed no differences between *Pkd2* mutants with or without *Jnk* (S5 Fig). We expect that *Pkd2* mutants would develop kidney cysts at older ages based on evidence from *Pkd1* models [52], but severe liver findings (Fig 7) precluded further aging.

Polycystic liver disease is the most common extrarenal ADPKD symptom [54] Interestingly, despite lacking kidney cysts, adult *Pkd2* mutants contained numerous biliary liver cysts. Livers were enlarged and indurated, with visible fluid-filled cysts causing a yellowish hue (Fig 7B). Trichrome staining showed extensive cystic and fibrotic changes throughout, with rare areas of healthy tissue. We observed numerous small and occasional large cysts, frequently surrounded by collagen (Fig 7A). JNK inhibition significantly improved liver cysts, reducing liver to body weight ratio by 38% (Fig 7C). Polycystic liver disease is more prevalent in females than males [55], but all mice in our study developed liver cysts after *Pkd2* deletion with no difference between males and females (S5 Fig). JNK-deleted cystic livers were smaller and remained indurated and pale (Fig 7B) with reduced cysts and fibrosis (Fig 7A). In addition to driving cyst progression in juvenile *Pkd2* mutant kidneys, JNK signaling also contributes to cyst progression in adult *Pkd2* mutant livers.

## Discussion

Improving ADPKD treatment requires understanding signaling downstream of the polycystins. We investigated JNK's role in promoting cysts due to Pkd2 loss and found that disrupting JNK signaling reduced kidney cysts in juvenile mice and liver cysts in adult mice. Our findings invite further exploration of JNK as a therapeutic target for ADPKD. JNK inhibitors have successfully treated liver and kidney disease in animals [24–26,56–58]. Unfortunately, toxic effects ended multiple human clinical trials [59] suggesting that JNK inhibition would not be appropriate in chronic conditions like ADPKD. However, it is expected that the loss of Pkd2 would

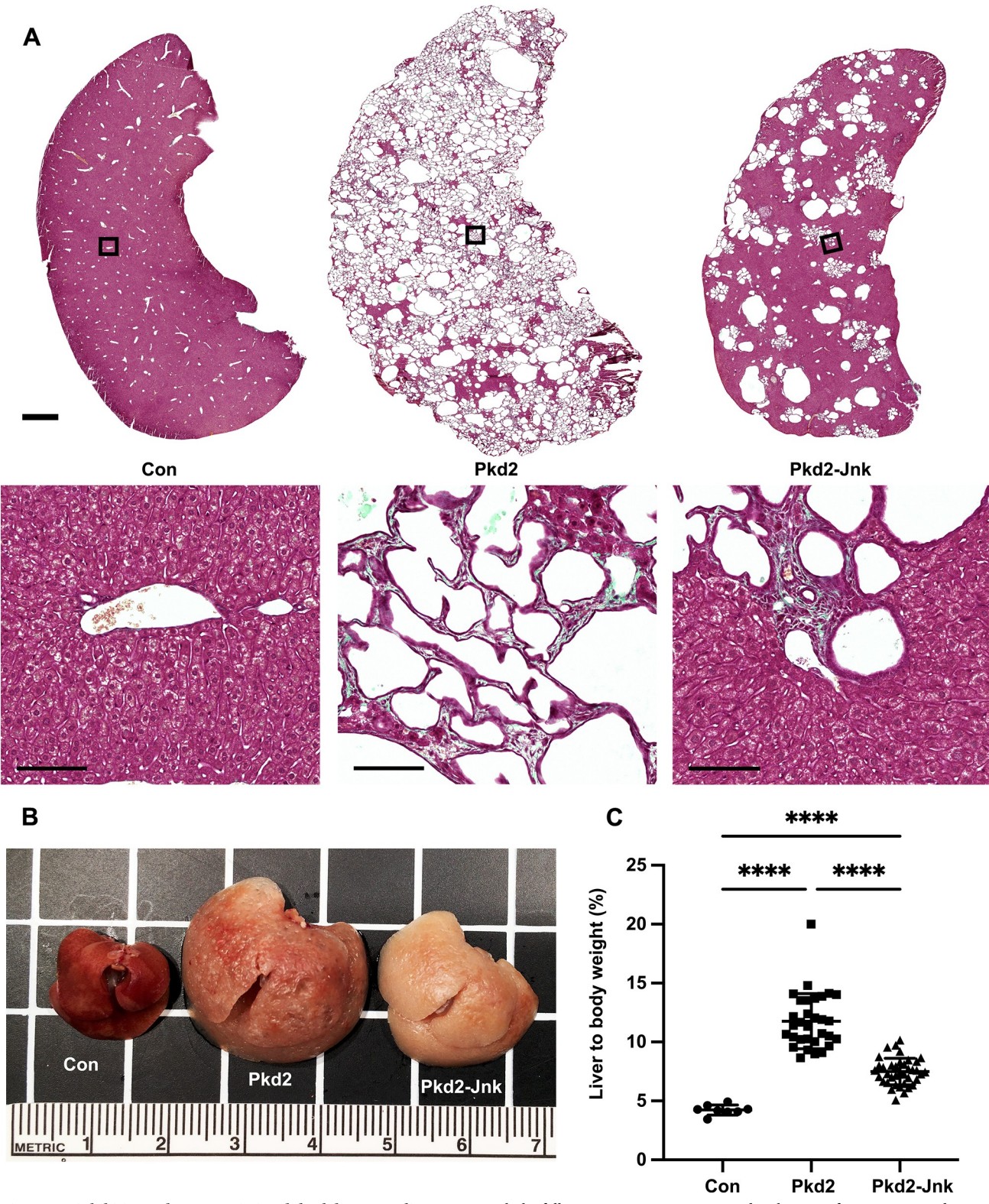

**Fig 7. JNK inhibition reduces cysts in in adult Pkd2 mutant livers.** Mice with the following genotypes were treated with tamoxifen at P21-23 and collected 24 weeks later: Con (Rosa26-CreERT2; Pkd2fl/+), Pkd2 (Rosa26-CreERT2; Pkd2fl/fl; Jnk1+/+, fl/+; Jnk2+/+, +/-), Pkd2-Jnk (Rosa26-CreERT2; Pkd2fl/fl; Jnk1fl/fl; Jnk2null/null). (A) Liver sections were stained with one-step trichrome to mark collagen fibers pale green, cytoplasm red, and nuclei dark blue. Black boxes indicate magnified regions. Scale bar is 1000 microns for full size liver scans, 100 microns for insets. (B) Gross morphology of

cystic livers. Centimeter ruler is shown. (C) Cystic burden in the liver was quantified using the ratio of liver weight / body weight x 100%. N is 8 (Con), 31 (Pkd2), 38 (Pkd2-Jnk). ****, P < 0.0001; ***, P < 0.001 by one-way ANOVA followed by Tukey multiple comparison test with multiplicity-adjusted p-values. Error bars indicate SD.

activate a MAP3K upstream of JNK. MAP3K inhibitors are in development as clinical reagents, and these do not show the toxic effects of Jnk inhibitors [60–64] making them more appropriate for ADPKD. Of course, identification of the relevant MAP3K is required. Mouse and human genomes encode 24 MAP3Ks, with at least 14 known to phosphorylate Mkk4 or Mkk7 upstream of Jnks [18,65] Our studies suggest that Ask1, Mlk2 and Mlk3 are not relevant, leaving the critical MAP3K still to be identified.

Understanding how the polycystin complex regulates JNK activity is an important question. MAPK signaling is typically activated by receptor tyrosine kinases, G-protein coupled receptors (GPCRs) and other membrane proteins detecting stress, cytokines, growth factors and other agonists. Polycystins can function as atypical GPCRs to activate heterotrimeric G proteins [13,15,66]. AP-1 and JNK activity are increased in HEK-293T cells upon transfection with membrane-targeted PKD1 constructs. Expression of dominant negative heterotrimeric G proteins abrogated the activation, while wild type G protein expression augmented it supporting a role for GPCRs and heterotrimeric G-proteins in JNK activation [13]. Alternatively, less direct mechanisms such as altered calcium signaling could be involved as calcium levels regulate JNK activity [67]. Polycystin-1 activates calcium influx after binding Wnt ligands [68] and polycystin mutations are known to reduce cellular calcium levels [5]. Signals are often propagated from the receptor to the MAP3Ks by the action of Rac1 and Cdc42, but other mechanisms are also used. For example, Ask1 is inhibited by binding thioredoxin, which disrupted by oxidative stress [69]. Work from Arnould *et al.* in cultured cells, showed that overexpression of Pkd2 or the C-terminal tail of Pkd1 can activate JNK and this was blocked by dominant negative mutations of Rac1 and Cdc42 [12,14] suggesting that signaling by small G proteins transmits the signal from the polycystins to Jnk. However, these experiments utilized over expression of the polycystins to activate JNK, while cystic disease is thought to be driven by reduced polycystin activity. Overexpression of the polycystins causes cyst formation, but it is not certain that the same mechanism drives cystic growth in both cases [70,71].

Gene expression profiles of cystic kidneys extensively overlap with kidney injury profiles [72]. In animal models, kidney injury activates JNK in the tubule epithelium and JNK inhibition prior to kidney injury reduces inflammation and fibrosis, and preserves kidney function [23–27]. Importantly, kidney injury exacerbates polycystic kidney disease [28–30]. Our data showing reduced cystic burden with JNK loss suggests that JNK is in the pathway that leads to cyst initiation or expansion. It is possible that JNK activation is a response to the kidney injury caused by cyst growth. Injury-induced JNK activation may exacerbate or perpetuate tubule cell proliferation in a "feed-forward" mechanism. Further investigation of this theory would include kidney injury in Pkd knockout mice. Supporting evidence would consist of attenuation of accelerated cyst growth in JNK-inhibited animals. Regardless, the observation that JNK loss/inhibition reduces the severity of disease caused by loss of polycystin-2 or kidney injury indicates that JNK is important to the pathology of cystic disease.

The protective effect of JNK inhibition on cystic disease is driven by *Jnk1*. Jnk1 and Jnk2 are structurally similar [50] and have overlapping functions. However, evidence suggests distinct and even opposing roles for Jnks in tissue homeostasis and cancer [73,74]. Functional differences between Jnk1 and Jnk2 may be due to alternative processing of their transcripts. Two independent splicing events produce four distinct isoforms of both genes. One event alters the open reading frame at the C-terminus [50] and has unknown consequences to protein

function. Another event dictates inclusion of mutually exclusive exons 7a/7b within the kinase domain and appears to affect kinase activity [75]. Jnk splicing in mouse kidney is uncharacterized, but the Jnk1 and Jnk2 produced in kidney may contain different versions of exon 6, which could influence isoform activity or substrate specificity.

94% of ADPKD patients develop hepatic cysts by their fourth decade, but most remain asymptomatic [76] Rarely, severe liver involvement requires surgery to reduce liver volume [55,77] In our adult model of *Pkd2* deletion, we found extensive liver cysts after six months, without detectable kidney cysts. Similarly, liver cysts were detected earlier than kidney cysts in an adult mouse model of Cre-mediated *Pkd1* deletion [52] In our model, *Jnk* deletion suppressed hepatic cysts. An important caveat to our results is that we did not test the effect of chronic Jnk inhibition in adult animals with functional Pkd2. A previous study demonstrated a requirement of *Jnk* for hepatic cyst formation in models of mitochondrial redox stress-induced cholangiocarcinoma [78] However, this role of JNK to promote cystogenesis appears to be dependent on physiological context because *Jnk* deletion alone was sufficient to cause hepatic cyst formation in another study of aged mice [79–81] JNK signaling may have tissue-specific roles in ADPKD that will be important to evaluate for therapeutic development.

Our work demonstrates that Pkd2 loss activates JNK and genetic removal of Jnk reduces cystic burden in mice with Pkd2 mutations. Further studies will elucidate the pathway by which polycystin loss leads to JNK dysregulation, but our work suggests that JNK pathway inhibitors should be explored as treatment for ADPKD.

## Supporting information

**S1 Fig. Variation in *Pkd2* mRNA levels is similar in *Pkd2* mutants with or without Jnk.** (A) Total RNA was extracted from P21 kidneys. *Pkd2* mRNA levels were determined by RT-qPCR and normalized to *Gapdh*. All animals were treated with tamoxifen by maternal oral gavage on P2-4. The groups consist of Cre-negative animals (N = 8), *Pkd2* heterozygotes with any number of Jnk alleles (*Rosa26-Cre$^{ERT2}$; Pkd2$^{fl/+}$*) (N = 5), *Pkd2* mutants with at least one wild type allele of *Jnk1* and *Jnk2* (*Rosa26-Cre$^{ERT2}$; Pkd2$^{fl/fl}$*) (N = 15), and *Pkd2* mutants with no functional *Jnk* alleles (*Rosa26-Cre$^{ERT2}$; Pkd2$^{fl/fl}$; Jnk1$^{fl/fl}$; Jnk2$^{null/null}$*) (N = 12). ****, P < 0.0001; ***, P < 0.001; *, <0.05; ns, not significant by one-way ANOVA followed by Tukey multiple comparison test with multiplicity-adjusted p-values. Error bars indicate SD. (B) Data from A replotted against 2-kidney to body weight. *Pkd2* mRNA levels were plotted against 2-kidney to body weight for individual Pkd2 mutants with at least one wild type allele of *Jnk1* and *Jnk2* (PKD2, N = 15) or lacking all functional *Jnk* alleles (PKD2-JNK, N = 12). Each group was analyzed by simple linear regression. For a given reduction in *Pkd2* mRNA, mutants lacking Jnk tend to have a lower cystic burden. However, the slopes of the two lines were not statistically significantly different. PKD2: Y = -0.1212*X + 1.240, PKD2-JNK: Y = -0.1593*X + 1.312. (TIF)

**S2 Fig. Deletion of MAP3 Kinases *Ask1*, *Mlk2* and *Mlk3* is not sufficient to reduce kidney cysts in juvenile *Pkd2* mutant mice.** Mice with the following genotypes were treated with tamoxifen at P2-4 and collected at P21: Con (*Rosa26-Cre$^{ERT2}$; Pkd2$^{fl/+}$*), Pkd2 (*Rosa26-Cre$^{ERT2}$; Pkd2$^{fl/fl}$*), Pkd2-Ask1 (*Rosa26-Cre$^{ERT2}$; Pkd2$^{fl/fl}$; Ask1$^{-/-}$*) and Pkd2-Mlk2/3 (*Rosa26-Cre$^{ERT2}$; Pkd2$^{fl/fl}$; Mlk2$^{-/-}$; Mlk3$^{-/-}$*). (A) Cystic burden was quantified using the ratio of 2-kidney weight / body weight x 100%. N is 17 (Con), 25 (Pkd2), 23 (Pkd2-Ask1). ****, P < 0.0001; ns, not significant by one-way ANOVA followed by Tukey multiple comparison test with multiplicity-adjusted p-values. Error bars indicate SD. (B) Cystic burden was quantified using the ratio of 2-kidney weight / bodyweight x 100%. N is 12 (Con), 23 (Pkd2), 11 (Pkd2-Mlk2/3). ****, P < 0.0001; ***, P < 0.001; ns, not significant by one-way ANOVA followed by Tukey

multiple comparison test with multiplicity-adjusted p-values. Error bars indicate SD.
(TIF)

**S3 Fig. *Pkd2* loss induces phosphorylation of long and short isoforms of *Jnk1* and *Jnk2*.**
Mice described in Fig 2: Con (*Rosa26-Cre$^{ERT2}$; Pkd2$^{fl/+}$*), Pkd2 (*Rosa26-Cre$^{ERT2}$; Pkd2$^{fl/fl}$; JNK1$^{+/+}$; Jnk2$^{+/+}$*), Pkd2-Jnk (*Rosa26-Cre$^{ERT2}$; Pkd2$^{fl/fl}$; Jnk1$^{fl/fl}$; Jnk2$^{null/null}$*) Mice described in
Fig 6: Pkd2-Jnk1 (*Rosa26-Cre$^{ERT2}$; Pkd2$^{fl/fl}$; Jnk1$^{null/null}$; Jnk2$^{+/+}$*); Pkd2-Jnk2 (*Rosa26-Cre$^{ERT2}$; Pkd2$^{fl/fl}$; JNK1$^{+/+}$; Jnk2$^{null/null}$*). (A) Whole kidney protein lysates were immunoblotted for total
Jnk, phospho T183/Y185 Jnk, and loading control Gapdh. The diagram (left) depicts the pri-
mary known isoforms of Jnk1 and Jnk2, as well as a putative isoform of Jnk1. Both Jnk1 and
Jnk2 can be alternatively spliced to form four variants. Incorporation of the mutually exclusive
exon pair 7a/7b in the kinase domain does not affect size of the protein, but alternative splicing
at the C-terminus can produce either long (p54) or short (p46) forms that can be distinguished
by immunoblot. Colored arrows indicate corresponding isoforms on the blot. These represent
our predictions of which isoforms are present in each band. The black arrows correspond to a
band of unknown identity (p45) which we quantified separately. (B-C) Quantification of
immunoblots described in (A). N is 3 animals per group. Due to low sample number, differ-
ences in total Jnk isoform levels did not reach statistical significance in most cases and are not
displayed. For phosphorylated Jnk isoforms, we indicate the significance of each group com-
pared to *Pkd2* mutants. Phospho-p45 nearly reached significance with p = 0.058. $^{**}$, P < 0.01;
$^{*}$, P < 0.05; ns, not significant by one-way ANOVA followed by Tukey multiple comparison
test with multiplicity-adjusted p-values. Error bars indicate SD. (D) The kidney lysates
described in (A) were immunoblotted for phospho T180/Y182 p38 and loading control
Gapdh.
(TIF)

**S4 Fig. Kidney cysts do not develop within 6 months of adult Pkd2 deletion.** Mice with the
following genotypes were treated with tamoxifen at P21-23 and collected 24 weeks later: Con
(*Rosa26-Cre$^{ERT2}$; Pkd2$^{fl/+}$*), Pkd2 (*Rosa26-Cre$^{ERT2}$; Pkd2$^{fl/fl}$; Jnk1$^{+/+, fl/+}$; Jnk2$^{+/+, +/-}$*), Pkd2-Jnk
(*Rosa26-Cre$^{ERT2}$; Pkd2$^{fl/fl}$; Jnk1$^{fl/fl}$; Jnk2$^{null/null}$*). (A) Kidney sections were stained with one-step
trichrome to mark collagen fibers pale green, cytoplasm red, and nuclei dark blue. Scale bar is
1000 microns for full size kidney scans, 100 microns for insets. (B) Cystic burden in the kidney
was quantified using the ratio of 2-kidney / body weight x 100%. N is 8 (Con), 31 (Pkd2), 38
(Pkd2-Jnk). ns, not significant by one-way ANOVA followed by Tukey multiple comparison
test with multiplicity-adjusted p-values. Error bars indicate SD.
(TIF)

**S5 Fig. Sex does not influence severity of cystic phenotype in adult *Pkd2* mutant mice.**
Mice with the following genotypes were treated with tamoxifen at P21-23 and collected 24
weeks later: Con (*Rosa26-Cre$^{ERT2}$; Pkd2$^{fl/+}$*), Pkd2 (*Rosa26-Cre$^{ERT2}$; Pkd2$^{fl/fl}$; Jnk1$^{+/+, fl/+}$; Jnk2$^{+/+, +/-}$*), Pkd2-Jnk (*Rosa26-Cre$^{ERT2}$; Pkd2$^{fl/fl}$; Jnk1$^{fl/fl}$; Jnk2$^{null/null}$*). (A) Body weight of
adult *Pkd2* mutant mice, segregated by sex. Females: N is 3 (Con), 15 (Pkd2), 19 (PKD2-JNK).
Males: N is 5 (Con), 16 (Pkd2), 19 (Pkd2-Jnk). $^{****}$, P < 0.0001; $^{**}$, P < 0.01; ns, not significant
by one-way ANOVA followed by Tukey multiple comparison test with multiplicity-adjusted
p-values. Error bars indicate SD. (B-C) 2-kidney weight and 2-kidney to body weight (%) of
adult *Pkd2* mutant mice, segregated by sex. Females: N is 3 (Con), 15 (Pkd2), 19 (PKD2-JNK).
Males: N is 5 (Con), 16 (Pkd2), 19 (Pkd2-Jnk). $^{****}$, P < 0.0001; $^{***}$, P < 0.001; $^{**}$, P < 0.01;
$^{*}$, P < 0.05; ns, not significant by one-way ANOVA followed by Tukey multiple comparison
test with multiplicity-adjusted p-values. Error bars indicate SD. (D-E) Liver weight and liver to
body weight (%) of adult *Pkd2* mutant mice, segregated by sex. Females: Females: N is 3 (Con),

15 (Pkd2), 19 (PKD2-JNK). Males: N is 5 (Con), 16 (Pkd2), 19 (Pkd2-Jnk). ****, P < 0.0001; **, P < 0.01; ns, not significant by one-way ANOVA followed by Tukey multiple comparison test with multiplicity-adjusted p-values. Error bars indicate SD.
(TIF)

## Acknowledgments

We thank Dr. Ichigo for providing *Ask1*$^{-/-}$ mice.

## Author Contributions

**Conceptualization:** Abigail O. Smith, Roger J. Davis, Gregory J. Pazour.

**Formal analysis:** Abigail O. Smith, Gregory J. Pazour.

**Funding acquisition:** Roger J. Davis, Gregory J. Pazour.

**Investigation:** Abigail O. Smith, Julie A. Jonassen, Kenley M. Preval, Gregory J. Pazour.

**Methodology:** Abigail O. Smith, Gregory J. Pazour.

**Resources:** Roger J. Davis.

**Supervision:** Gregory J. Pazour.

**Visualization:** Abigail O. Smith.

**Writing – original draft:** Abigail O. Smith, Gregory J. Pazour.

**Writing – review & editing:** Abigail O. Smith, Julie A. Jonassen, Kenley M. Preval, Roger J. Davis, Gregory J. Pazour.

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
