## [Decision Letter · Decision Letter 0]

24 Aug 2021

Dear Dr Pazour,

Thank you very much for submitting your Research Article entitled 'c-Jun N-terminal kinase (JNK) signaling contributes to cystic burden in polycystic kidney disease' to PLOS Genetics.

The manuscript was fully evaluated at the editorial level and by independent peer reviewers. The reviewers were generally favorable, but raised some concerns about the current manuscript. Based on the reviews, we will not be able to accept this version of the manuscript, but we would be willing to review a revised version. We cannot, of course, promise publication at that time.

If you decide to revise the manuscript for further consideration at PLOS Genetics, please aim to resubmit within the next 60 days, unless it will take extra time to address the concerns of the reviewers, in which case we would appreciate an expected resubmission date by email to plosgenetics@plos.org.

[LINK]

We are sorry that we cannot be more positive about your manuscript at this stage. Please do not hesitate to contact us if you have any concerns or questions.

Yours sincerely,

David R. Beier

Associate Editor

PLOS Genetics

Gregory Barsh

Editor-in-Chief

PLOS Genetics

Reviewer's Responses to Questions

**Comments to the Authors:**

Reviewer #1: In this manuscript the authors show convincingly that inactivation of JNK1/2 ameliorates kidney and liver cyst formation in Pkd2 mutants. The genetic data is convincing, with compound genetic inactivation of the JNKs in the Pkd2 mutant background, and the genetic analysis includes sample numbers adequate for interpretation. Positive aspects include the rigor of the genetic analysis and the quantitative metrics used. The major negative is that there is no specific upstream link between JNKs and Pkd2. As described in the discussion, this will be the subject of continued research, and the authors have made a significant effort to exclude 3 upstream kinases through compound genetic analyses. This negative data is not presented and to convincingly exclude these kinases the authors should include a table showing sample numbers analyzed for each. This genetic analysis provides new insight and should be of strong interest in the PKD field. Some specific comments on the data:

1. To look for candidate kinases upstream of JNK, why not use the same approach with the Pkd2 mutant lysates as used for JNK shown in figure 1?

2. Jun phosphorylation looks like it is limited to distended collecting ducts - could the authors investigate if this is the case and propose an explanation? One would expect activation of SAPK in any distended tubules I think, and data presented in the figure questions the causality of Jun phosphorylation in the cyst formation event.

3. The myofibroblast staining is intriguing, but it is not informative to compare terminally damaged Pkd2 mutant tissue with significantly less damaged Pkd2-Jnk tissue if the goal is to ask if JNK specifically affects fibrosis in the Pkd2 mutant. To get at any effect of JNK inactivation on myofibroblast abundance, tissues with the same degree of damage would need to be compared. Perhaps the simplest way to do this would be to show data from 3 or 4 time points P2-P21 and compare tissues with similar cystic indices.

4. The data in Figure 6A is difficult to interpret because the colors are hard to distinguish.

5. The interpretation of the banding pattern of the JNK and pJNK immunoblots is quite important to the results section but the data is not shown in the main figures and the interpretation is not very clear. The conclusion from the genetic analysis is that JNK1 is the predominant actor in cyst formation, and one assumes that the focus of the immunoblot analysis is on finding JNK1-specific molecular mechanisms. Oddly, deletion of JNK1 appears to have a far more modest effect on JNK phosphorylation in the Pkd2 mutant than deletion of JNK2. However, it looks like there are 3 bands above the 38kD marker in the Pkd2 lane and that the middle one is missing in Pkd1-Jnk and Pkd2-Jnk1 but the interpretation of this data needs to be clarified.

Reviewer #2: Synopsis;

This report demonstrates via genetic means the role of JNK, a terminal kinase in MAPK signaling, in controlling cystic burden in a recessive mouse model of human autosomal dominant polycystic kidney disease. Conditional deletion of the PKD2 gene during early post-natal life (P2-P3) results in a rapid kidney cyst growth and loss of kidney function. Concomitant conditional deletion of JNK1 and JNK2 genes dramatically reduces, but does not eliminate, cystic growth. Moreover, the authors demonstrate that JNK1 is sufficient to elicit this ameliorative phenotype. While JNK inhibitors have not been successfully deployed as a therapeutic strategy, upstream inhibition (MAPKKK) could provide an approach. Notably the authors were unable to identify an upstream kinase that might represent a point for therapeutic intervention. This would require further study.

General Comments:

The genetic approach is well done and the detailed measurements of genotype and JNK and c-jun phosphorylation provide critical support for a proposed epistatic cascade. The cascade as presented, would leave the reader with a view that JNK signaling is initiated directly downstream of PKD signaling and that abating that signaling cascade results in a reduced proliferative rate and reduced cystic burden. The authors also point out the role of JNK1 in a cuter kidney injury and the role of acute kidney injury in the acceleration of cystic progression. This inter-relationship between PKD signaling, cystic growth and injury and JNK signaling paints a more nuanced picture of the field, one that requires deeper dissection than is developed in this report. The authors can provide more nuanced picture as part of their discussion. One that might outline the complex interplay making it important to understand the the role JNK signaling with respect to PKD vs. cyst-induced injury and potential positive feedback between these cascades. The authors might even propose some directions that could distinguish between the many signaling pathways (e.g. greater temporal resolution in cystic onset and JNK signaling).

This report provide critical evidence to support a deeper dissection of the detailed mechanism and providing that more nuanced view brings out the richness and complexity of the underlying questions. Ones that can drive the field forward with greater clarity than simple epistatic relationships.

Specific Comments:

1. In discussing treatment of the early post-natal animals via maternal tamoxifen administration, using the term “indirect” or reminding the reader of the maternal transfer is appropriate at each mention of the P2/P3 condition treatment regimen.

2. FIgure 1: The presented immunoblot does not support the idea that the level of JNK is similar in the cystic kidney as the WT kidney. While the level of increase in p-JNK (and p-c-jun) is far greater than the level of JNK increase, an increase in JNK levels could be interesting either due to increased proliferation of tubule cells or even due to inflammatory infiltrates seen in high cystic burden. One remedy would be to provide, as supplementary data, the entire set of immunoblots to understand the variability in JNK levels, or an example with more representative JNK levels.

3. Also worth pointing out that the Rosa-CRE-ER approach does result in systemic deletion and that tubular cell autonomous mechanisms have been proposed, there remains the possibility that other cell types, such as inflammatory or immune cells may play a role. Conditional models with greater tissue specificity would provide a view on this.

Reviewer #3: In this study, Smith et al. investigate the role of Jnk signaling in an ADPKD mouse model. ADPKD is a very frequent genetic disease leading to end-stage kidney failure, typically in adults. So far, there is no effective therapy for patients except for the V2R antagonist tolvaptan that comes with massive side effects. Thus, identifying novel targets to interfere with the progression of this genetic disorder is very important.

To study JNK in ADPKD, the authors use a genetic mouse model which carries a floxed Pkd2 allele combined with tamoxifen-inducible (ubiquitous) Cre (in R26). They cross this model with a floxed Jnk1 allele and with a conventional Jnk2 knockout. In brief, the authors show that double knockout of Jnk1/2 reduces the severity of the PKD phenotype while inhibiting phospho-c-Jun accumulation in nuclei of tubular epithelial cells and reducing cell proliferation. In addition, they show that loss of Jnk1 alone is almost as efficient as the double knockout, and they also observe an amelioration of the liver phenotype in this Pkd2 model (here with late induction). Taken together, this is an important in vivo study that suggests a novel therapeutic target for a frequent and significant disease. The manuscript is very well written, and I really like the story. However, I have some concerns that should be addressed by the authors.

1. The genetic background of the mouse lines should be described. Is this all pure BL6, SV129, FVB, … or mixed? A mixed background could be critical and responsible for some of the more subtle effects observed.

2. Control animals are always referred to as “Wild Type” or “Wt” in the figures (starting with Fig 1A), but these animals are in fact heterozygous for Pkd2 and perhaps additional alleles. I feel that this should be more accurate, in particular, since ADPKD is an autosomal-dominant disease. Moreover, the genotype of control animals is not always clear to me.

3. Fig.2 describes the effect of Jnk1/2 deficiency on the PKD phenotype. In this figure, the phenotype of the Jnk1/2 double knockout without PKD2 deletion should be shown as an additional control. The same should be added to Fig. 7 (liver; in particular given the paper of Muller et al (Ref 75).

While the positive effect of Jnk1/2 deletion on renal cyst development is very promising, this effect could be in part the consequence of less efficient Pkd2 deletion by cre in presence of an additional floxed allele (Jnk1). The authors observe a huge variability both in the kidney-to-bodyweight ratio and in Pkd2 expression levels (Fig S1). It might be helpful to correlate Pkd2 expression with the cystic phenotype on the level of the individual animals.

In Fig S1: Please note, that the numbers from the figure legend (N=12 and N=15) do not fit with the number of data points in the graph. Here, either data sets or numbers were interchanged.

4. When investigating the impact of the individual Jnk kinases, jnk1 and Jnk2, the authors use a total body knockout of Jnk1 as an additional model (Fig. 6). Since the three panels on the left and the right panel of the figure refer to the previously used mouse model, it would be more convincing to see the phenotype of “R26-CreErt2; Pkd2fl/fl; Jnk1fl/fl;Jnk2+/+” here. It does not become evident why the authors now switch to the Jnk1-/- line instead.

Minor point:

1. I would like to suggest being somewhat more careful at the end of the discussion when suggesting: “JNK pathway inhibitors could be effective treatments for ADPKD”. This conclusion is currently based on one mouse model. Many other treatments for ADPKD do work in mice but not in humans (e.g., rapamycin). In addition, loss of Jnk1/2 has been recently shown to cause liver cysts in mice, indicating that general inhibition of Jnk kinases might not be straightforward. Please remember that ADPKD patients typically know a lot about their disease, and they follow the literature...

2. Fig. 3, Fig. 4 and Fig. 5 could also be combined to one figure.

**Have all data underlying the figures and results presented in the manuscript been provided?**

Reviewer #1: Yes

Reviewer #2: **No: **Was unable to find access to all the datasets (e.g. immunoblots) that supported the graphical representation. Particularly when variation is high, access to all immunoblots, rather than representative can be instructive.

Reviewer #3: Yes

PLOS authors have the option to publish the peer review history of their article (what does this mean?). If published, this will include your full peer review and any attached files.

Reviewer #1: No

Reviewer #2: No

Reviewer #3: No

---

## [Decision Letter · Decision Letter 1]

2 Dec 2021

Dear Dr Pazour,

Thank you very much for submitting your Research Article entitled 'c-Jun N-terminal kinase (JNK) signaling contributes to cystic burden in polycystic kidney disease' to PLOS Genetics.

The manuscript was fully evaluated at the editorial level and by independent peer reviewers. Reviewer #3 identified a minor concern which we ask you to address. Your response will be assessed editorially; that is, the manuscript will not be sent for further review

We therefore ask you to modify the manuscript according to the review recommendations. Your revisions should address the specific points made by each reviewer.

[LINK]

Yours sincerely,

David R. Beier

Associate Editor

PLOS Genetics

Gregory Barsh

Editor-in-Chief

PLOS Genetics

Reviewer's Responses to Questions

**Comments to the Authors:**

Reviewer #1: All questions have been answered.

Reviewer #2: Have address key concerns of this reviewer.

Look forward to follow-up mechanistic studies and therapeutic implications.

Reviewer #3: The authors addressed most of my concerns. I apologize that I did not understand the "making-of" the Jnk1 null/null line correctly. Of course, the authors are right, no problem in using this - based on the very same allele.

I am still unhappy with the label "Wt" in several figures refering to heterozygous knockout animals.

In addition, Fig. S6 and S7 should be either rearranged or deleted.

**Have all data underlying the figures and results presented in the manuscript been provided?**

Reviewer #1: Yes

Reviewer #2: Yes

Reviewer #3: Yes

PLOS authors have the option to publish the peer review history of their article (what does this mean?). If published, this will include your full peer review and any attached files.

Reviewer #1: No

Reviewer #2: No

Reviewer #3: No

---

## [Editor Report · Decision Letter 2]

11 Dec 2021

Dear Dr Pazour,

We are pleased to inform you that your manuscript entitled "c-Jun N-terminal kinase (JNK) signaling contributes to cystic burden in polycystic kidney disease" has been editorially accepted for publication in PLOS Genetics. Congratulations!

Yours sincerely,

David R. Beier

Associate Editor

PLOS Genetics

Gregory Barsh

Editor-in-Chief

PLOS Genetics

Comments from the reviewers (if applicable):

**Data Deposition**

http://datadryad.org/submit?journalID=pgenetics&manu=PGENETICS-D-21-00927R2

**Press Queries**

---

## [Editor Report · Acceptance letter]

21 Dec 2021

PGENETICS-D-21-00927R2 

c-Jun N-terminal kinase (JNK) signaling contributes to cystic burden in polycystic kidney disease 

Dear Dr Pazour, 

We are pleased to inform you that your manuscript entitled "c-Jun N-terminal kinase (JNK) signaling contributes to cystic burden in polycystic kidney disease" has been formally accepted for publication in PLOS Genetics! Your manuscript is now with our production department and you will be notified of the publication date in due course.

With kind regards,

Zsofia Freund

PLOS Genetics

On behalf of:
